# Self-Supervised Representation Learning As Mutual Information Maximization

## Abstract

Self-supervised representation learning (SSRL) has demonstrated remarkable empirical success, yet its underlying principles remain insufficiently understood. While recent works attempt to unify SSRL methods by examining their information-theoretic objectives or summarizing their heuristics for preventing representation collapse, architectural elements like the predictor network, stop-gradient operation, and statistical regularizer are often viewed as empirically motivated additions. In this paper, we adopt a first-principles approach and investigate whether the learning objective of an SSRL algorithm dictates its possible optimization strategies and model design choices. In particular, by starting from a variational mutual information (MI) lower bound, we derive two training paradigms, namely Self-Distillation MI (SDMI) and Joint MI (JMI), each imposing distinct structural constraints and covering a set of existing SSRL algorithms. SDMI inherently requires alternating optimization, making stop-gradient operations theoretically essential. In contrast, JMI admits joint optimization through symmetric architectures without such components. Under the proposed formulation, predictor networks in SDMI and statistical regularizers in JMI emerge as tractable surrogates for the MI objective. We show that many existing SSRL methods are specific instances or approximations of these two paradigms. This paper provides a theoretical explanation behind the choices of different architectural components of existing SSRL methods, beyond heuristic conveniences.

## 1 Introduction

SSRL has achieved significant success by learning useful features from unlabeled data, achieving competitive performance with supervised approaches across a wide range of tasks (LeCun et al., 2015; Bengio et al., 2013; Balestriero et al., 2023). Conventionally, SSRL algorithms can be divided into two categories according to their training objectives, contrastive methods and non-contrastive methods. Contrastive methods (Oord et al., 2018; Tian et al., 2020; Chen et al., 2020a; He et al., 2020; Chen et al., 2020b; 2021) train a representation model by aligning the representations of augmentations of the same input while explicitly pushing apart representations of augmentations of different inputs. On the other hand, non-contrastive methods (Grill et al., 2020; Chen & He, 2021; Caron et al., 2021; Zbontar et al., 2021; Bardes et al., 2022; Sui et al., 2024) challenge the necessity of negative samples and propose alternative mechanisms, such as the use of momentum encoders or stop-gradient operations to prevent representational collapse.

Many recent studies have attempted to unify these two categories of SSRL methods under common theoretical frameworks, often through shared information-theoretic principles. Liu et al. (2022) interpreted various SSRL methods as low-order approximations of a unified maximum entropy principle; Zbontar et al. (2021) applied Information Bottleneck theory (Tishby et al., 1999; Tishby & Zaslavsky, 2015) to explain the Barlow Twins objective, while Tsai et al. (2021) later linked it to a kernel-based MI measure; Shwartz-Ziv et al. (2023) linked VICReg's penalties to MI bounds; and most recently, Jha et al. (2024) proposed a unifying framework that explains collapse avoidance based on minimizing a global mean while preserving augmentation-level variation. Despite their insights, prior work offers little clarity on whether training strategies like self-distillation or variance–covariance control are heuristic additions or principled consequences of the objective itself, leaving an important theoretical gap in understanding.

In this work, we bridge the gap by returning to first principles, grounding our analysis of existing SSRL algorithms through the lens of MI maximization, a shared underlying objective of almost all self-supervised learning methods. Starting from a variational lower bound on MI, particularly the Donsker-Varadhan (DV) bound, we show that it naturally leads to two optimization paradigms in the context of SSRL: Self-Distillation MI (SDMI), which uses EM-style alternating updates with stop-gradient operations (e.g., SimSiam, BYOL, MoCo), and Joint MI (JMI), which supports joint optimization via a single gradient step per batch (e.g., SimCLR, Barlow Twins, VICReg). More specifically, we note that dividing SSRL algorithms based on this new taxonomy is theoretically more principled than the traditional contrastive vs. non-contrastive distinction. In addition, based on the SDMI and JMI paradigms, we further generalize these paradigms into canonical algorithmic forms, and demonstrate that they behave similarly to existing SSRL methods in the corresponding paradigms and can achieve competitive performance on downstream tasks.

In summary, our contributions are as follows:

1. We formulate a general MI maximization perspective under the DV bound, showing that existing SSRL methods implicitly follow one of two optimization paradigms, namely Self-Distillation MI (SDMI) or Joint MI (JMI).

2. We show that design elements like stop-gradients, exponential moving average targets, predictor networks, and statistical regularizers are not heuristics, but theoretically necessary under MI-based objectives, providing a formal explanation for common design choices.

3. We show that many well-known SSRL approaches (e.g., SimCLR, BYOL, SimSiam) can be mapped directly to our two paradigms. This helps unify the field under a shared theoretical lens and offers guidance for future method design.

## 2 RELATED WORK AND PRELIMINARIES

We begin by reviewing recent attempts to unify the growing landscape of SSRL methods under shared theoretical principles. We first summarize key unification efforts based on objective design and collapse-prevention mechanisms, highlighting their contributions and limitations. We then present MI as a foundational concept and starting point for our analysis, revisiting its definition and variational lower bounds, with a focus on the DV bound whose tightness and decomposition are central to our work.

### 2.1 UNIFICATION APPROACHES IN SSRL

A growing body of work (Zbontar et al., 2021; Liu et al., 2022; Tsai et al., 2021; Shwartz-Ziv et al., 2023; Jha et al., 2024; Tan et al., 2024) suggests that an information-theoretic lens can help unify seemingly disparate SSRL methods. Many existing methods, particularly contrastive approaches, can be explicitly framed as maximizing MI between representations of different augmented views (Oord et al., 2018; He et al., 2020; Chen et al., 2020b; 2021; Poole et al., 2019).

Building on this, several works have linked specific SSRL losses to MI estimation. The Information Bottleneck perspective (Tishby et al., 1999; Tishby & Zaslavsky, 2015) has been applied to Barlow Twins (Zbontar et al., 2021), and Tsai et al. (2021) showed that the Barlow Twins objective is equivalent to maximizing a Hilbert–Schmidt Independence Criterion (Gretton et al., 2005), a kernelized dependence measure related to MI. Bardes et al. (2022) introduced VICReg's variance and covariance penalties, and Shwartz-Ziv et al. (2023) later provided an information-theoretic analysis linking these penalties to MI bounds.

Another prominent unification direction is offered by Liu et al. (2022), who propose a Maximum Entropy Coding (MEC) framework that treats representation learning as an entropy maximization problem, showing that many existing SSRL methods can be interpreted as low-order Taylor approximations of a single entropy-based objective. Complementing this view, Jha et al. (2024) analyze the collapse avoidance mechanisms that ensure stability in SSRL, arguing that, despite architectural and algorithmic differences, most methods implicitly minimize the global average of learned representations while preserving sample-level variability.

While these approaches provide valuable unifying perspectives on SSRL objectives or collapse-prevention mechanisms, they do not address whether the commonly used optimization strategies

and architectural components, such as stop-gradient operations, predictor networks, or statistical regularizers, are necessary consequences of the learning objective itself or simply heuristic choices.

## 2.2 Mutual information and its variational bounds in SSRL

In SSRL, MI is often defined between representations $Z_A$ and $Z_B$ of two augmented views $X_A$ and $X_B$ of an input $X$ in the form $I(Z_A; Z_B) = D_{\mathrm{KL}}[p(z_A, z_B) \| p(z_A)p(z_B)]$. Maximizing MI with respect to the encoding function $Z_A = f_\theta(X_A)$ defines a valid pretext task for learning representations that can transfer to various downstream applications. However, direct optimization of MI is intractable since the underlying data distribution $P(X)$ is unknown, motivating the use of variational bounds in practice. Common variational bounds include InfoNCE (Oord et al., 2018; Poole et al., 2019), Barber–Agakov (Barber & Agakov, 2003), TUBA (Poole et al., 2019), NWJ (Nguyen et al., 2010), JSD (Hjelm et al., 2019) and DV (Belghazi et al., 2018). Each of these alternatives introduces different trade-offs between tightness, stability, and optimization feasibility.

We use the Donsker–Varadhan (DV) bound to guide our analysis in this paper, as it offers: (1) a direct connection to MI via KL divergence, (2) a natural variational decomposition that facilitates block-coordinate ascent, and (3) is provably tighter than $f$-divergence-based alternatives for any fixed function class (Belghazi et al., 2018).

**Donsker-Varadhan bound**  Over a sufficiently rich class of functions $\mathcal{F}$, the DV bound decomposes MI as:

$$I(Z_A; Z_B) \geq I_{\mathrm{DV}}(Z_A; Z_B) = \sup_{T \in \mathcal{F}} \left\{ \underbrace{\mathbb{E}_{p(z_A, z_B)}[T(z_A, z_B)]}_{\text{Joint term}} - \underbrace{\log \mathbb{E}_{p(z_A)p(z_B)}\left[e^{T(z_A, z_B)}\right]}_{\text{Marginal term}} \right\}, \quad (1)$$

where $\mathcal{F} \subseteq \{ f : \mathscr{Z}_A \times \mathscr{Z}_B \to \mathbb{R} \}$, while $T \in \mathcal{F}$ is a scoring function that assigns high values to joint pairs $(z_A, z_B) \sim p(z_A, z_B)$ and low values to marginal pairs $(z_A, z_B) \sim p(z_A)p(z_B)$.

## 3 A Unified View of SSRL as MI Maximization

In this section, we first revisit the DV lower bound on MI from an optimization perspective. This gives rise to two natural optimization paradigms in SSRL, namely Self-Distillation MI (SDMI) and Joint MI (JMI), respectively. Then, we analytically show how a wide range of SSRL methods can be categorized under these paradigms.

### 3.1 Block-coordinate ascent via DV bound

Let representations $Z_A$ and $Z_B$ come from two different encoding functions $f_\theta$ and $g_\xi$ with a fixed scoring function $T$ drawn from the function class $\mathcal{F}$. We note the DV bound shown in eq. (1) provides a useful formulation for optimization since exact maximization of the bound with respect to the encoder parameters $\theta$ for view $Z_A$ while holding $\xi$ for $Z_B$ fixed, and vice-versa guarantees a non-decreasing improvement of the objective. As a result, alternating updates over the encoders for $Z_A$ and $Z_B$ constitute valid block-coordinate ascent steps. Specifically, we can formalize the improvement as follows:

**Proposition**  Let the DV-bound objective be given by

$$\mathcal{L}(\theta, \xi) = J(\theta; \xi) - M(\theta; \xi), \quad (2)$$

where $\mathcal{L}(\theta, \xi)$ is the DV bound, $J(\theta; \xi)$ is the joint term, and $M(\theta; \xi)$ is the marginal term from eq. (1). Assume that: (1) for fixed $\xi$, $J(\cdot; \xi)$ is concave in $\theta$; (2) the marginal term $M(\cdot; \xi)$ is smooth and satisfies $\|\nabla_\theta M(\theta; \xi)\| \leq \varepsilon$; and (3) the same conditions hold symmetrically for updates over $\xi$. Then alternating gradient steps over $\theta$ and $\xi$ yields approximate monotonic improvement in $\mathcal{L}(\theta, \xi)$:

$$\mathcal{L}(\theta^{(k+1)}, \xi^{(k)}) \geq \mathcal{L}(\theta^{(k)}, \xi^{(k)}) - \mathcal{O}(\varepsilon), \quad \mathcal{L}(\theta^{(k+1)}, \xi^{(k+1)}) \geq \mathcal{L}(\theta^{(k+1)}, \xi^{(k)}) - \mathcal{O}(\varepsilon).$$

See Section A.1 for our proof. In particular, if $\varepsilon \to 0$ (e.g., slowly changing marginal distributions), the objective becomes asymptotically non-decreasing over iterations.

When sharing parameters $\theta = \xi$, the maximization objective in eq. (2) can be jointly optimized via standard gradient ascent with the guarantee of monotonic improvement, provided that the full

objective $\mathcal{L}(\theta)$ is concave. In the case of the DV bound (eq. (1)), this holds because the joint term is concave and the marginal term is convex, making the overall objective concave.

As such, there are two valid optimization paradigms to maximize MI: alternating updates across encoder branches or joint updates over shared parameters. We name the two paradigms Self-Distillation MI (SDMI) and Joint MI (JMI), respectively.

## 3.2 SELF-DISTILLATION MUTUAL INFORMATION (SDMI)

SSRL methods in the SDMI paradigm rely on an EM-style alternating update schedule between two encoder branches and a mechanism for maximizing MI between augmented views. The alternating updates are enabled through a stop-gradient operator, which breaks the gradient flow from one branch to the other, making it possible to treat one encoder as fixed while updating the other, mimicking a block-coordinate ascent on the DV bound. Typically, these methods use an online encoder that receives direct gradient updates and a target (or momentum) encoder that is updated via an exponential moving average (EMA) of the online encoder's parameters. While some existing SDMI methods such as SimSiam and BYOL do not explicitly optimize a variational MI bound, we show that their alternating update structure, enabled by stop-gradients and architectural asymmetry, can be derived as a principled optimization strategy for DV-bound maximization. This provides a theoretical justification for previously heuristic design choices.

**Block-coordinate interpretation of SDMI** To formalize SDMI as an EM-style block-coordinate ascent procedure, we consider batches of two augmented views $X_1 = \{x_1^i\}_{i=1}^N$ and $X_2 = \{x_2^i\}_{i=1}^N$, where each $x_1^i, x_2^i$ is sampled from a stochastic augmentation $\mathcal{A}(x)$ applied to an input $x \sim P(x)$ with batch size $N$, and two encoders $f_\theta$ and $g_\xi$.

**E-Step:** At iteration $k$, we define the MI between the representations produced by the encoders $f_\theta$ and $g_\xi$ as

$$I^{(k)} = I\big(f_{\theta^{(k)}}(X_1), \ g_{\xi^{(k)}}(X_2)\big). \tag{3}$$

We update the $f_\theta$ encoder by maximizing the objective under a stop-gradient (SG) on the $g_\xi$ encoder:

$$\theta^{(k+1)} = \arg\max_\theta \ I\big(f_\theta(X_1), \ \mathrm{SG}(g_{\xi^{(k)}}(X_2))\big) \tag{4}$$

which guarantees $\quad I\big(f_{\theta^{(k+1)}}(X_1); \ g_{\xi^{(k)}}(X_2)\big) \geq I\big(f_{\theta^{(k)}}(X_1); \ g_{\xi^{(k)}}(X_2)\big).$

**M-Step:** Using the updated $f_\theta$ encoder, we update the $g_\xi$ encoder with a stop-gradient on $f_\theta$,

$$\xi^{(k+1)} = \arg\max_\xi \ I\big(\mathrm{SG}(f_{\theta^{(k+1)}}(X_1)), \ g_\xi(X_2)\big), \tag{5}$$

ensuring $\quad I\big(f_{\theta^{(k+1)}}(X_1); \ g_{\xi^{(k+1)}}(X_2)\big) \geq I\big(f_{\theta^{(k+1)}}(X_1); \ g_{\xi^{(k)}}(X_2)\big).$

**Monotonic Improvement:** Together, these steps guarantee overall monotonic improvement:

$$I\big(f_{\theta^{(k+1)}}(X_1), \ g_{\xi^{(k+1)}}(X_2)\big) \geq I\big(f_{\theta^{(k+1)}}(X_1), \ g_{\xi^{(k)}}(X_2)\big) \geq I\big(f_{\theta^{(k)}}(X_1), \ g_{\xi^{(k)}}(X_2)\big) \tag{6}$$

### 3.2.1 EXAMPLES OF SDMI METHODS

**SimSiam and BYOL** Implicit contrastive methods, such as BYOL (Grill et al., 2020) and SimSiam (Chen & He, 2021), fall under the SDMI paradigm. These methods train an online encoder $f_\theta$, together with a lightweight predictor $h_\phi$, to align transformed representations with those of a target encoder $g_\xi$. From the SDMI viewpoint, both methods approximate a two-step EM-style optimization in a relaxed, implicit form:

1. **E-step:** In the E-step, both methods update the online encoder to maximize MI by minimizing the following negative cosine similarity loss

$$\min_{\theta,\phi} \ - \mathbb{E}_{p(x_1,x_2)}\big[T_{\cos}\big(h_\phi(f_\theta(x_1)), \ g_\xi(x_2)\big)\big], \tag{7}$$

where $T_{\cos}$ denotes a cosine similarity scoring function. This loss can be viewed as an instantiation of the DV bound with cosine similarity, which we refer to as $I_{\cos\text{-DV}}$. However, these methods omit the explicit marginal term present in the full bound (see eq. (14)), relying instead on their predictor dynamics to discourage collapse.

2. **M-step (Implicit):** Immediately after the E-step, SimSiam resets the target encoder with the new online weights and freezes it for the next E-step:

$$g_{\text{new}} = \text{SG}(f_\theta). \tag{8}$$

BYOL, on the other hand, uses an EMA of $\theta$:

$$\xi \leftarrow \tau\xi + (1 - \tau)\theta. \tag{9}$$

While these methods differ from SDMI's explicit coordinate ascent step on the $g_\xi$ encoder, they preserve the underlying principle of alternating optimization, though in an implicit form.

Our interpretation aligns with the hypothesis of Chen & He (2021) that SimSiam's stop-gradient induces EM-like alternating updates between online and frozen branches. While they suggested that the predictor approximates an expectation over augmentations, Zhang et al. (2022) refuted this, showing instead that it induces de-centering and de-correlation gradients that stabilize training and promote feature diversity. Within our SDMI framework, we reinterpret these effects as implicitly approximating the marginal term of the DV bound. Section D.2.2 provides further analysis and empirical evidence in support of this interpretation.

**MoCo**  MoCo (He et al., 2020; Chen et al., 2020b; 2021), a contrastive learning method, also fits naturally within the SDMI paradigm. It performs EM-style alternating updates between an online encoder and a momentum encoder, while directly optimizing the InfoNCE lower bound on MI. Its momentum encoder plays a similar functional role and is updated via EMA, like the target encoder in BYOL. Early versions (MoCo-v1 (He et al., 2020), v2 (Chen et al., 2020b)) already achieve strong performance without predictor networks, and although MoCo-v3 (Chen et al., 2021) introduces a predictor, it yields only marginal performance gains ($\sim 1\%$), underscoring that with direct MI maximization, predictors are auxiliary.

This illustrates how the SDMI framework unifies both traditional contrastive and non-contrastive methods under a shared lens of MI maximization with alternating encoder updates.

### 3.3  JOINT MUTUAL INFORMATION (JMI)

Unlike SDMI, JMI methods use a single encoder $f_\theta$ to produce representations for both augmented views, enabling joint gradient updates to maximize MI. It is achieved either by directly optimizing an explicit MI objective or by incorporating surrogate regularization terms that penalize statistical properties, such as variance, covariance, or feature redundancy, to approximate the marginal log-partition term in eq. (1). A general JMI objective written as a loss function takes the form

$$\mathcal{L}_{\text{JMI}} = -\mathbb{E}_{p(x_1, x_2)}\left[T(f_\theta(x_1), f_\theta(x_2))\right] + \lambda \cdot \mathcal{R}(f_\theta(x_1), f_\theta(x_2)). \tag{10}$$

Examples of JMI methods include contrastive learning methods such as SimCLR, which directly optimizes InfoNCE to maximize MI between views. More recent implicit contrastive methods, such as Barlow Twins (Zbontar et al., 2021) and VICReg (Bardes et al., 2022), optimize an alignment term between augmented views and a regularizer that approximates the marginal term from eq. (1).

### 3.4  FROM DV TO BARLOW TWINS: A SURROGATE DERIVATION

To show how implicit contrastive methods can be seen as using feature-level regularization as in eq. (10), we demonstrate how the Barlow Twins loss could be derived from eq. (1) using several straightforward approximations and assumptions, providing a direct connection of the Barlow Twins loss to mutual information maximization between views. To begin, we replace the DV bound's marginal term with its second order Taylor approximation:

$$\mathcal{L}_{\text{Taylor-DV}} = -\underbrace{\mathbb{E}_{p(z_A, z_B)}[T(z_A, z_B)]}_{\text{Joint term}} + \underbrace{\mathbb{E}_{p(z_A)p(z_B)}[T(z_A, z_B)]}_{\text{Marginal mean term}} + \underbrace{\text{Var}_{p(z_A)p(z_B)}[T(z_A, z_B)]}_{\text{Marginal variance term}}. \tag{11}$$

Barlow Twins corresponds to the particular choice of the dot product scoring function,

$$T(z^A, z^B) = \sum_{i=1}^{d} z_i^A z_i^B, \tag{12}$$

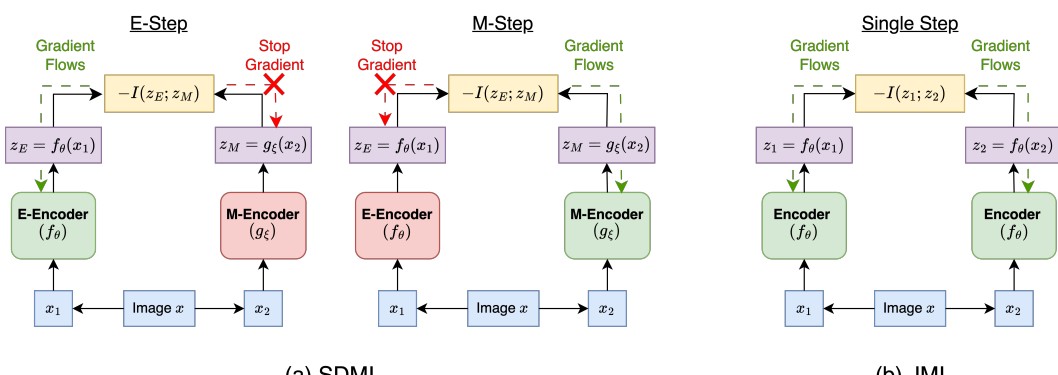

(a) SDMI                                                          (b) JMI

Figure 1: Canonical forms of our proposed paradigms: (a) SDMI alternates updates between two encoders using stop-gradients, while (b) JMI jointly updates both views with shared gradients.

which is an approximation to the optimal $T$ in eq. (1). Since batch normalization is normally applied, we also assume $\mathbb{E}[z_i^A] \approx \mathbb{E}[z_i^B] \approx 0$, which effectively removes the marginal mean term from eq. (11), leaving the alignment and variance terms as the primary components. Barlow Twins is usually expressed with the empirical cross-correlation matrix,

$$C_{ij} = \frac{1}{N} \sum_{n=1}^{N} z_{n,i}^A \cdot z_{n,j}^B,$$

where alignment is encouraged via the diagonal $C_{ii}$, and redundancy is penalized via the off-diagonals $C_{ij}$, $i \neq j$. To simplify the variance term in eq. (11), we expand it using the dot product in eq. (12), and further assume jointly Gaussian representations with decorrelation within each view, which then implies (by Isserlis' theorem (Munthe-Kaas et al., 2025))

$$\mathrm{Var}\left[ \sum_i z_i^A z_i^B \right] \approx \sum_{i \neq j} C_{ij}^2.$$

Putting together these components yields a moment-based surrogate to the DV objective

$$\mathcal{L}_{\text{Taylor-DV}} \approx -\sum_i C_{ii} + \sum_{i \neq j} C_{ij}^2, \tag{13}$$

which closely matches the Barlow Twins loss. We provide the full derivation in section A.3.

In summary, SDMI and JMI represent two principled optimization paradigms for maximizing MI. Our findings reveal that many architectural components in modern SSRL methods, previously introduced as heuristic choices, can instead be interpreted as structured consequences of optimizing MI. We illustrate the distinction between SDMI and JMI in fig. 1, and give in section C a summary of representative SSRL methods and their classification under the SDMI/JMI taxonomy, including whether they employ explicit MI objectives or surrogate regularizers.

## 4 EXPERIMENTS

This section empirically validates the theoretical structure of SDMI and JMI by instantiating their canonical forms and analyzing their behavior alongside representative SSRL methods. The purpose of this study is not to suggest the canonical forms of SDMI and JMI are state-of-the-art SSRL methods. Instead, we use them as a simplified setting to understand the dynamics of MI training, representation quality, and to examine how the optimization principles derived from MI manifest in practice. We compare the canonical forms to more specialized and performant variants from the literature to shed light on the role of MI maximization in SSRL.

### 4.1 CANONICAL SDMI AND JMI PROTOTYPES

To empirically validate the theoretical framework developed in section 3, we instantiate minimal, controlled implementations of the SDMI and JMI paradigms. These *canonical forms* exclude auxiliary components such as momentum updates, predictor networks, or regularizers, and serve to isolate

the optimization structure derived from the DV bound. As illustrated in fig. 1, SDMI alternates updates between two encoders using stop-gradients, while JMI applies symmetric joint updates to both augmented views using a shared encoder. Both prototypes optimize the same MI objective defined below, enabling a direct comparison of their dynamics.

**Objective: Cosine-based DV bound**    While the DV bound is theoretically maximally tight when $\mathcal{F}$ is a sufficiently broad class of functions, in practice, unrestricted neural critics $T$ often lead to high variance and unstable training behavior (Oord et al., 2018; Poole et al., 2019; Song & Ermon, 2020). To ensure reliable estimation while preserving the validity of DV bound, we restrict the critic function $T$ to cosine similarity, $T(z_A, z_B) = \frac{z_A \cdot z_B}{\|z_A\|_2 \|z_B\|_2}$, providing a stable, bounded, and scale-invariant surrogate. This choice is further motivated by its widespread use in SSRL objectives (Chen et al., 2020a; He et al., 2020; Chen et al., 2020b; 2021; Grill et al., 2020; Chen & He, 2021), where it serves as a standard metric for comparing representations across augmented views. By restricting $T$ to be the cosine similarity, we effectively optimize only over the representations of $Z_A$ and $Z_B$:

$$I(Z_A; Z_B) \geq I_{\mathrm{DV}}(Z_A; Z_B) \geq I_{\text{cos-DV}}(Z_A; Z_B)$$

$$= \mathbb{E}_{p(z_A, z_B)}\left[T_{\cos}(z_A, z_B)\right] - \log \mathbb{E}_{p(z_A)p(z_B)}\left[e^{T_{\cos}(z_A, z_B)}\right]. \quad (14)$$

Although using $I_{\text{cos-DV}}$ sacrifices some tightness, it provides a more stable estimator while remaining a lower bound of the MI objective.

**Practical approximation**    To compute the marginal term in eq. (14) efficiently, we approximate the expectation using off-diagonal cross-pairs from a batch of size $N$:

$$\log \mathbb{E}_{P(z_A)P(z_B)}\left[e^{T_{\cos}(z_A, z_B)}\right] \approx \log\left(\frac{1}{N(N-1)} \sum_{\substack{i,j=1 \\ i \neq j}}^{N} e^{T_{\cos}(z_A^{(i)}, z_B^{(j)})}\right).$$

Hence, our batchwise training objective takes the form:

$$\mathcal{L}_{\text{cos-DV}} = -\left[\frac{1}{N} \sum_{i=1}^{N} T_{\cos}(z_A^{(i)}, z_B^{(i)}) - \log\left(\frac{1}{N(N-1)} \sum_{\substack{i,j=1 \\ i \neq j}}^{N} e^{T_{\cos}(z_A^{(i)}, z_B^{(j)})}\right)\right]. \quad (15)$$

### 4.2 EXPERIMENTAL SETUP

**Datasets**    We utilize standard datasets used for SSRL tasks including CIFAR10/100 (Krizhevsky & Hinton, 2009), TinyImageNet, and ImageNet100 (Deng et al., 2009). Additionally, for controlled experiments and visualization, we generate a toy dataset from a mixture of five isotropic Gaussian distributions centered at evenly spaced points on the unit circle. Each cluster center is defined by $\mu_k = \left(\cos\frac{2\pi k}{5}, \sin\frac{2\pi k}{5}\right)$, $k = 1, \ldots, 5$, with samples drawn as $x \sim \mathcal{N}(\mu_k, \sigma^2 I)$, where $\sigma = 0.05$ and $I$ is the $2 \times 2$ identity matrix. Two augmented views are generated by perturbing $x$ with independent Gaussian noise: $x_1 = x + \epsilon_1$, $x_2 = x + \epsilon_2$, $\epsilon_1, \epsilon_2 \sim \mathcal{N}(0, \tau^2 I)$, where $\tau = 0.1$. We generate $N = 2500$ samples, with $n_{\text{per\_cluster}} = 500$ per class.

**Implementation details**    We implement the canonical SDMI prototype (fig. 1(a)) with two independently initialized encoders trained via alternating E- and M-step updates, while the JMI prototype (fig. 1(b)) uses a single shared encoder updated jointly with symmetric gradients, and all baseline methods use their standard architectures and objectives. Our canonical SDMI and JMI prototypes use ResNet-18 (He et al., 2016) encoders for CIFAR10/100 and TinyImageNet, and ResNet-50 encoders for ImageNet100 (Deng et al., 2009). We use a smaller network for the Gaussian dataset, described in section E.4.

**Mutual information estimation**    To assess MI dynamics during training, we compute three variational bounds: the cos-DV bound ($I_{\text{cos-DV}}$) from eq. (14), the InfoNCE bound ($I_{\text{InfoNCE}}$) (Oord et al., 2018; Poole et al., 2019), and the JSD bound ($I_{\text{JSD}}$) (Hjelm et al., 2019).

For JMI-based methods (JMI prototype, SimCLR, BarlowTwins and VICReg), both augmented views are passed through the same encoder $f_\theta$, and MI is computed between the representations:

$$I^{(t)} = I\big(f_\theta^{(t)}(x_1), f_\theta^{(t)}(x_2)\big).$$

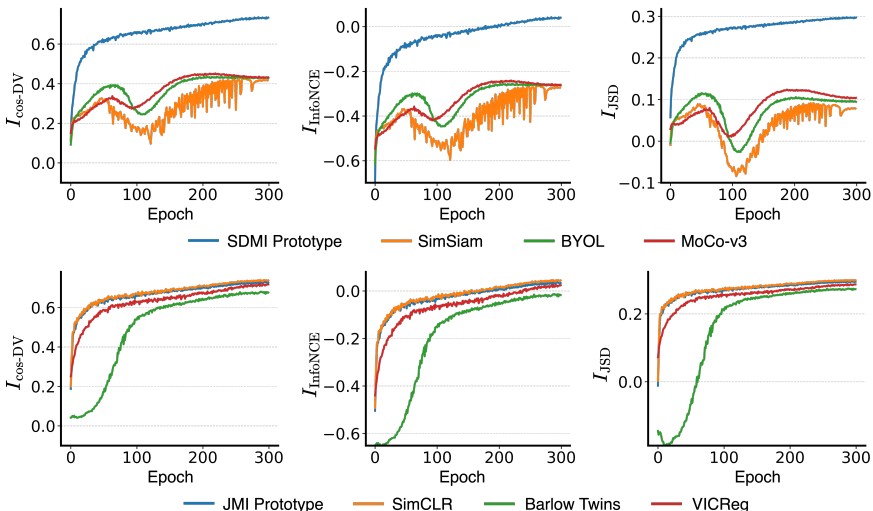

Figure 2: Estimated MI over CIFAR10 training for SDMI-based (top row) and JMI-based (bottom row) methods, using three estimators (cos–DV, InfoNCE and JSD; left to right). Both paradigms exhibit consistent MI growth: SDMI curves feature early fluctuations before trending upward, while JMI estimates rise more uniformly, and to much higher levels.

For SDMI-based methods (SDMI prototype, SimSiam, BYOL, MoCo-v3), MI is measured between two asymmetric encoder branches. In the SDMI prototype, these are independently updated $f_\theta$ and $g_\xi$ encoders trained via alternating updates:

$$I^{(t)} = I\big(f_\theta^{(t)}(x_1),\, g_\xi^{(t)}(x_2)\big).$$

In BYOL and MoCo-v3, $g_\xi$ is a momentum encoder updated via EMA. In SimSiam, which lacks a persistent target encoder, we instead treat the previous epoch's encoder state as the M-branch:

$$I^{(t)}_{\text{SimSiam}} = I\big(f_\theta^{(t)}(x_1),\, f_\theta^{(t-1)}(x_2)\big), \text{with } I^{(0)} = -\infty \text{ by convention.}$$

### 4.3  RESULTS

**Monotonic MI increase**  Figure 2 shows estimated MI over training for all methods across both paradigms on the CIFAR10 dataset. Since the SDMI prototype explicitly optimizes the cos–DV bound in eq. (14), while MoCo-v3 optimizes InfoNCE, the JSD bound serves as an independent estimator not optimized by any method. Compared to the other SDMI methods, the SDMI prototype (top row) exhibits a near-perfect monotonic increase in MI throughout training. This is expected, as it explicitly optimizes the cos-DV bound (eq. (14)) using true EM-style alternating updates between two independently parameterized encoders. In contrast, methods like SimSiam, BYOL, and MoCo only approximate this behavior through their architectural heuristics, which leads to a noisy MI estimate and generally lower final MI levels. Nevertheless, all methods still exhibit an overall upward MI trend, confirming that they retain the underlying MI-maximization structure. Meanwhile, all JMI-based methods display smooth and consistently increasing MI curves, reflecting their symmetric joint-update optimization. We provide additional results on the Gaussian data in section D.2, confirming this trend in ideal conditions.

**Cluster center trajectories in embedding space**  To visualize how well the representation space separates underlying structure, we use the Gaussian dataset and track the movement of all five cluster centers during training in fig. 3. We quantify separation via the nearest-neighbor (NN) angle gap, the mean angular distance to the closest other center. The SDMI prototype achieves the largest separation, with an average nearest-neighbor (NN) angle gap of $\approx 77°$, compared to $\approx 54°$ of the strongest analogous methods. Detailed metrics and comparative analysis are presented in section D.1.

**Linear probing**  To assess the quality of learned representations for downstream tasks, we perform linear probing on real-world datasets. We trained encoders, then froze them to train a linear classifier head using cross-entropy (Tian et al., 2020). As shown in table 1, our prototype methods are competitive with established SSRL methods across both SDMI and JMI paradigms. No single method outperforms all others consistently. See section E for implementation details.

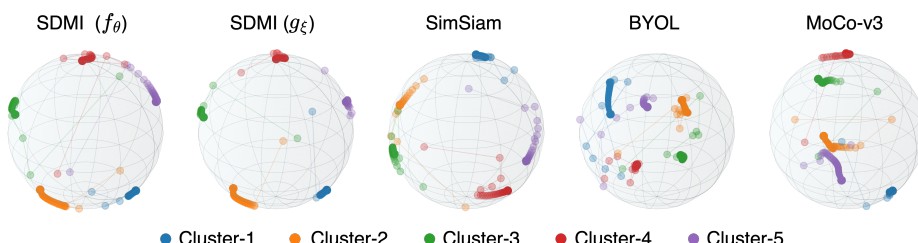

Figure 3: Embedding trajectories of the five Gaussian cluster centers. Opacity increases over training. SDMI separates centers more distinctly than analogous methods.

Table 1: Linear probing accuracy (%) on four datasets. Mean ± std over 3 runs.

| Model | CIFAR10 | CIFAR100 | TinyImageNet | ImageNet100 |
|---|---|---|---|---|
| SDMI prototype ($f_\theta$) | 88.61 $\pm 0.13$ | 57.37 $\pm 0.38$ | 33.30 $\pm 0.58$ | 70.73 $\pm 0.57$ |
| SDMI prototype ($g_\xi$) | 88.59 $\pm 0.35$ | 57.85 $\pm 0.32$ | 32.94 $\pm 0.71$ | 70.83 $\pm 0.16$ |
| SimSiam | 89.72 $\pm 0.18$ | 60.45 $\pm 0.60$ | 19.19 $\pm 0.69$ | 78.23 $\pm 0.58$ |
| BYOL | 91.28 $\pm 0.16$ | 63.11 $\pm 0.21$ | 32.77 $\pm 0.10$ | 81.09 $\pm 0.61$ |
| MoCo-v3 | 91.10 $\pm 0.16$ | 58.90 $\pm 0.32$ | 32.18 $\pm 0.55$ | 76.86 $\pm 0.74$ |
| JMI Prototype | 88.01 $\pm 0.48$ | 57.22 $\pm 0.56$ | 32.23 $\pm 0.52$ | 73.41 $\pm 0.36$ |
| SimCLR | 87.24 $\pm 0.37$ | 55.32 $\pm 0.46$ | 33.79 $\pm 0.31$ | 75.31 $\pm 0.76$ |
| Barlow Twins | 85.56 $\pm 0.71$ | 51.91 $\pm 0.49$ | 30.26 $\pm 0.12$ | 78.96 $\pm 0.30$ |
| VICReg | 85.49 $\pm 1.03$ | 54.00 $\pm 0.34$ | 32.03 $\pm 0.32$ | 78.86 $\pm 0.23$ |

It is worth noting that our canonical SDMI and JMI models are intentionally minimal, showing that theory-driven models can provide strong baselines without the need for empirically-driven architectural tweaks like predictor heads, EMA, or regularization. Existing SSRL methods build on these baselines with architectural improvements. Our work focuses on explanation, and not optimization.

**Discussion** Interestingly, while the SDMI prototype achieves the highest MI under all three bounds, and the most separated clusters of representations, this does not translate directly into higher downstream performance. This suggests that maximizing MI, though necessary to prevent representational collapse, is not by itself sufficient for optimal SSRL performance. MI should thus be viewed as a foundation rather than the ultimate objective of SSRL. Crucially, our results show that the optimization paradigm, SDMI or JMI, and the strategies and components it uses, determine how the MI objective is approximated and, in turn, the usefulness of the learned features for downstream tasks.

In summary, 'how' MI is optimized matters as much as 'how much' MI is achieved. By formalizing the SDMI and JMI paradigms and identifying their essential components, our taxonomy provides a roadmap for future research. We recommend that future efforts prioritize designing better strategies and architectural components tailored to each optimization paradigm, thereby better bridging the gap between MI maximization and downstream task performance.

## 5 CONCLUSION

In this work, we revisited SSRL from first principles, grounding our analysis in a variational MI optimization lens. By deriving two distinct training paradigms, SDMI and JMI, we showed that many design choices in SSRL architectures are not merely empirical conveniences but theoretically motivated necessities. By unifying a broad class of existing SSRL methods under the theoretical lens, our analysis offers an alternative understanding of the mechanisms that drive successful representation learning and guides the principled design of future SSRL algorithms.

**Limitations** While our framework offers a principled view of SSRL via MI maximization, our experiments are limited to image datasets. Extending the analysis to other data modalities such as text, audio, or multimodal settings would strengthen the generalizability of our theoretical insights and is a promising direction for future work.

**Broader impact** By clarifying the principles behind self-supervised learning, this work may support more robust and interpretable model design. Though theoretical, our findings could influence the development of trustworthy AI systems in socially impactful domains.

## 6 REPRODUCIBILITY STATEMENT

We have made significant efforts to ensure that our results are fully reproducible. Section 3 formally derives the proposed SDMI and JMI paradigms and lists all assumptions, with complete proofs in section A and training procedures in section B. Section 4.2 describes our experimental setup, including synthetic data generation and evaluation protocols, while section E provides full implementation details, compute resources, random seed settings, and hyperparameter configurations for CIFAR10/100, TinyImageNet, and ImageNet100. Hyperparameter sweeps and selected settings are reported in table 4–table 7, and additional results and ablations are presented in section D. An anonymized implementation containing all code for model training, evaluation, and MI estimation is provided in the supplementary material to facilitate exact replication.

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

# A  FURTHER ANALYSIS

## A.1  BLOCK-COORDINATE ASCENT IN MI BOUNDS

We provide the formal proof for the proposition stated in section 3.1, establishing the theoretical foundation for monotonic MI increase under alternating optimization in the SDMI paradigm.

**Proof.**  Fix $\xi^{(k)}$. Since $J(\cdot; \xi^{(k)})$ is concave, a gradient ascent step on $\theta$ guarantees

$$J(\theta^{(k+1)}; \xi^{(k)}) \geq J(\theta^{(k)}; \xi^{(k)}). \tag{16}$$

By smoothness of $M$ and the bound $\|\nabla_\theta M\| \leq \varepsilon$, we have

$$\left| M(\theta^{(k+1)}; \xi^{(k)}) - M(\theta^{(k)}; \xi^{(k)}) \right| = \mathcal{O}(\varepsilon), \tag{17}$$

which yields

$$\mathcal{L}(\theta^{(k+1)}, \xi^{(k)}) \geq \mathcal{L}(\theta^{(k)}, \xi^{(k)}) - \mathcal{O}(\varepsilon). \tag{18}$$

An identical argument applies to the $\xi$-update. Chaining the two completes the proof.

## A.2  ANALYZING OTHER VARIATIONAL BOUNDS

Extending the analysis from section 3.1, we examine other commonly used variational MI bounds in SSRL, including InfoNCE and JSD bounds mentioned in section 2.2, demonstrating that our framework generalizes beyond the DV bound.

### A.2.1  INFONCE

Recall that the InfoNCE loss between two representations $Z_A$ and $Z_B$ takes the form:

$$
\begin{aligned}
\mathcal{L}_{\text{InfoNCE}} &= -\mathbb{E}_{p(z_A, z_B)} \left[ \log \left( \frac{e^{T(z_A, z_B)}}{\sum_{z'_B} e^{T(z_A, z'_B)}} \right) \right] \\
&= -\mathbb{E}_{p(z_A, z_B)} \left[ T(z_A, z_B) - \log \left( \sum_{z'_B} e^{T(z_A, z'_B)} \right) \right],
\end{aligned} \tag{19}
$$

where $T$ is a similarity function.

This loss can be interpreted as a lower bound on MI between $Z_A$ and $Z_B$ (Poole et al., 2019), such that:

$$I_{\text{InfoNCE}}(Z_A; Z_B) = \mathbb{E}_{p(z_A, z_B)} \left[ T(z_A, z_B) \right] - \mathbb{E}_{p(z_A)} \left[ \log \mathbb{E}_{p(z_B)} \left[ e^{T(z_A, z_B)} \right] \right] + \log N, \tag{20}$$

where $N$ is the number of negative samples.

Both the DV and the InfoNCE bounds follow the general structure: a joint term minus a marginal term. The only structural difference is that the DV bound aggregates globally before applying the logarithm:

$$\log \mathbb{E}_{p(z_A)p(z_B)} \left[ e^{T(z_A, z_B)} \right] \quad \text{(global aggregation)},$$

whereas InfoNCE applies the logarithm per sample:

$$\mathbb{E}_{p(z_A)}\left[\log \mathbb{E}_{p(z_B)}\left[e^{T(z_A, z_B)}\right]\right] \quad \text{(local aggregation).}$$

Although this difference affects the aggregation structure, both objectives satisfy the conditions of the proposition in eq. (2) and support monotonic improvement under alternating optimization.

### A.2.2 JSD

Similarly, the JSD bound can be expressed as:

$$I_{\text{JSD}}(Z_A; Z_B) = \mathbb{E}_{p(z_A, z_B)}\left[-\log\left(1 + e^{-T(z_A, z_B)}\right)\right] - \mathbb{E}_{p(z_A)p(z_B)}\left[\log\left(1 + e^{T(z_A, z_B)}\right)\right]$$

This form corresponds to a binary classification objective, distinguishing samples from the joint distribution versus the product of marginals. As with DV and InfoNCE, it has a "joint term minus marginal term" structure, but instead of a log-sum-exp aggregation, it applies the softplus nonlinearity independently to each sample.

All three objectives (DV, InfoNCE, JSD) satisfy the conditions of the proposition in eq. (2) and allow monotonic improvement under alternating optimization.

### A.3 MOMENT-BASED SURROGATES FOR THE DV MARGINAL TERM

We present the complete mathematical derivation referenced in section 3.4 showing that the regularizers in Barlow Twins correspond to a second-order Taylor expansion (cumulant expansion) of the DV bound's marginal term.

#### DV BOUND

To ground our approximation, we recall the DV bound (eq. (1)):

$$I(Z_A; Z_B) \geq I_{\text{DV}}(Z_A; Z_B) = \sup_{T \in \mathcal{F}} \left\{ \underbrace{\mathbb{E}_{p(z_A, z_B)}[T(z_A, z_B)]}_{\text{Joint term}} - \underbrace{\log \mathbb{E}_{p(z_A)p(z_B)}\left[e^{T(z_A, z_B)}\right]}_{\text{Marginal term}} \right\},$$

where $T \in \mathcal{F}$ is a critic function, chosen from a sufficiently expressive function class $\mathcal{F}$.

#### CGF AND TAYLOR EXPANSION

Let $T$ be any bounded critic with

$$T(x, y) \in [a, b] \quad \text{for all } x, y,$$

and define its CGF as

$$K(s) = \log \mathbb{E}[e^{sT}], \tag{21}$$

Because $T$ is bounded, $K$ is infinitely differentiable on $[0, 1]$, its $n$-th derivative at zero yields the $n$-th cumulant:

$$\kappa_n = K^{(n)}(0).$$

In particular,

$$K'(0) = \mathbb{E}[T], \quad K''(0) = \text{Var}(T).$$

By Taylor's theorem about $s = 0$, for $s \in [0, 1]$,

$$K(s) = s\,K'(0) + \tfrac{1}{2}s^2\,K''(0) + R_2(s), \tag{22}$$

where $R_2(s) = \frac{1}{6}s^3 K^{(3)}(\xi)$ for some $\xi \in (0, s)$.

Since $T \in [a, b]$, all derivatives of $K(s)$ are bounded on $[0, 1]$. In particular, evaluating eq. (22) at $s = 1$ yields:

$$\left| R_2(1) \right| \leq \frac{1}{6} \max_{s \in [0,1]} \left| K^{(3)}(s) \right| = \mathcal{O}(1).$$

This constant can therefore be absorbed into a hyperparameter. Hence, the second-order approximation holds in full generality:

$$\log \mathbb{E}[e^T] = K(1) \approx \mathbb{E}[T] + \tfrac{1}{2} \mathrm{Var}(T). \tag{23}$$

SURROGATE LOSS VIA MEAN-VARIANCE

Substituting eq. (23) into the DV bound (eq. (1)) gives the surrogate MI lower bound

$$\begin{aligned} I_{\mathrm{DV}}(Z_A; Z_B) \; &\geq \; I_{\text{Taylor-DV}}(Z_A; Z_B) \\ &= \mathbb{E}_{p(z_A, z_B)}[T(z_A, z_B)] - \left\{ \mathbb{E}_{p(z_A)p(z_B)}[T(z_A, z_B)] + \tfrac{1}{2} \mathrm{Var}_{p(z_A)p(z_B)}\big[T(z_A, z_B)\big] \right\}. \end{aligned} \tag{24}$$

Thus one may construct a tractable loss as shown in eq. (11)

$$\mathcal{L}_{\text{Taylor-DV}} = - \underbrace{\mathbb{E}_{p(z_A, z_B)}[T(z_A, z_B)]}_{\text{Joint term}} + \underbrace{\mathbb{E}_{p(z_A)p(z_B)}[T(z_A, z_B)]}_{\text{Marginal mean term}} + \underbrace{\mathrm{Var}_{p(z_A)p(z_B)}[T(z_A, z_B)]}_{\text{Marginal variance term}} {}^{1}.$$

BARLOW TWINS AS A MEAN–VARIANCE SURROGATE

We start with eq. (12):

$$X_i = z_i^A z_i^B, \quad T_{\cos}(z^A, z^B) = \sum_{i=1}^{d} X_i = z^A \cdot z^B$$

By the variance-of-a-sum identity,

$$\mathrm{Var}\big[T(z^A, z^B)\big] = \mathrm{Var}\Big( \sum_{i=1}^{d} X_i \Big) = \sum_{i,j=1}^{d} \mathrm{Cov}(X_i, X_j) = \sum_{i,j=1}^{d} \mathrm{Cov}(z_i^A z_i^B, \, z_j^A z_j^B). \tag{25}$$

Barlow Twins reduces the surrogate in eq. (11) to an alignment term and a tractable approximation of the marginal variance by applying batch normalization, ensuring

$$\mathbb{E}_{p(z_A)}[z_i^A] = \mathbb{E}_{p(z_B)}[z_i^B] \approx 0 \quad \Rightarrow \quad \mathbb{E}_{p(z_A)p(z_B)}[T(z_A, z_B)] \approx 0.$$

We write $z_i^A$ and $z_i^B$ to denote the $i$-th coordinate of views $A$ and $B$, respectively.

The regularization terms in Barlow Twins are constructed using batch-level statistics, specifically, the *cross-correlation matrix* between features across the two views:

$$C_{ij} = \frac{1}{N} \sum_{n=1}^{N} z_{n,i}^A \cdot z_{n,j}^B,$$

where $z_{n,i}^A$ and $z_{n,j}^B$ denote the $i$-th and $j$-th features of the $n$-th sample from each view in a batch of size $N$. The diagonal elements $C_{ii}$ appear in the alignment term of the loss, encouraging each feature to match across views, while the off-diagonal elements $C_{ij}$ for $i \neq j$ are penalized to reduce redundancy.

To connect this to the variance term in eq. (11), we analyze the variance of eq. (12) under independent sampling:

---

[1]The $\frac{1}{2}$ coefficient is omitted for simplicity, as it can be absorbed into a tuning hyperparameter.

$$\text{Var}_{p(x)p(y)}\left[\sum_{i=1}^{d} z_i^A z_i^B\right] = \sum_{i,j} \text{Cov}(z_i^A z_i^B, \ z_j^A z_j^B). \tag{26}$$

This covariance approximates a fourth-order moment:

$$\text{Cov}(X_i, X_j) = \mathbb{E}[z_i^A z_i^B z_j^A z_j^B] - \mathbb{E}[z_i^A z_i^B] \cdot \mathbb{E}[z_j^A z_j^B]. \tag{27}$$

Assuming that the representations are approximately jointly Gaussian and decorrelated within each view (i.e., $\mathbb{E}[z_i^A z_j^A] \approx 0$, $\mathbb{E}[z_i^B z_j^B] \approx 0$ for $i \neq j$), we can apply Isserlis' theorem (Munthe-Kaas et al., 2025) to approximate the fourth-order covariance terms:

$$\text{Cov}(z_i^A z_i^B, \ z_j^A z_j^B) \approx \mathbb{E}[z_i^A z_j^B] \cdot \mathbb{E}[z_j^A z_i^B] = C_{ij} C_{ji} \approx C_{ij}^2, \quad \text{for } i \neq j. \tag{28}$$

The variance thus approximates the sum of off-diagonal squared correlations:

$$\text{Var}\left[\sum_i z_i^A z_i^B\right] \approx \sum_{i \neq j} C_{ij}^2.$$

Putting everything together, the Taylor–DV surrogate yields eq. (13):

$$\mathcal{L}_{\text{Taylor-DV}} \approx -\sum_{i=1}^{d} C_{ii} + \sum_{i \neq j} C_{ij}^2,$$

which matches the structure of the empirical Barlow Twins loss: an alignment term encouraging the diagonal of the cross-correlation matrix to approach 1, and a decorrelation term penalizing off-diagonal elements.

### A.4 INPUT INFORMATIVENESS

While the SDMI and JMI frameworks increase $I(z_E; z_M)$, their effectiveness depends on how this relates to the input $x$. We formalize this intuition with the following conjecture.

**Conjecture.** Under the assumption of deterministic encoders, the MI between two distinct augmented views $z^{(1)}$ and $z^{(2)}$ is upper bounded by:

$$I(z^{(1)}; z^{(2)}) \leq \min(I(x; z^{(1)}); I(x; z^{(2)})) \tag{29}$$

**Proof.** Recall that MI between two random variables $A$ and $B$ is defined as:

$$I(A; B) = H(A) - H(A \mid B) = H(B) - H(B \mid A).$$

Since $z^{(1)} = f(x_1)$ and $z^{(2)} = f(x_2)$ (or $z^{(2)} = g(x_2)$ for SDMI) are deterministic functions of $x$, we have

$$H(z^{(1)} \mid x) = 0, \quad H(z^{(2)} \mid x) = 0.$$

Thus,

$$I(x; z^{(1)}) = H(z^{(1)}), \quad I(x; z^{(2)}) = H(z^{(2)}).$$

By definition,

$$I(z^{(1)}; z^{(2)}) = H(z^{(1)}) - H(z^{(1)} \mid z^{(2)}) \leq H(z^{(1)}),$$

where the inequality follows from the non-negativity of conditional entropy, $H(z^{(1)} \mid z^{(2)}) \geq 0$. Therefore,

$$I(z^{(1)}; z^{(2)}) \leq I(x; z^{(1)}).$$

By symmetry, we also have $I(z^{(1)}; z^{(2)}) \leq I(x; z^{(2)})$. Combining these gives

$$I(z_E; z^{(2)}) \leq \min\big\{I(x; z^{(1)}), I(x; z^{(2)})\big\}.$$

# B  ALGORITHMS

Detailed algorithmic descriptions for the canonical SDMI and JMI prototypes introduced in section 4.1 and illustrated in fig. 1 are provided below. Following common SSRL practice (Chen et al., 2020a; 2021; Grill et al., 2020; Chen & He, 2021), we adopt a symmetric loss by computing the objective over both view orderings.

## B.1  SDMI CANONICAL FORM TRAINING PROCEDURE

---

**Algorithm 1:** EM-style Training Procedure of the SDMI Prototype

---

**Input:** Unlabeled dataset $\mathcal{D}$, encoders $f_\theta$, $g_\xi$, temperature $\tau$, number of epochs $T$
**Output:** Trained encoder parameters $\theta$, $\xi$

1 **for** $t = 1$ **to** $T$ **do**
2   // E-Step: Update $f_\theta$, freeze $g_\xi$ **foreach** *minibatch* $(X_1, X_2) \sim \mathcal{D}$ **do**
2    $Z_E^{(1)} \leftarrow f_\theta(X_1)$,   $Z_E^{(2)} \leftarrow f_\theta(X_2)$
3    $Z_M^{(1)} \leftarrow g_\xi(X_1)$,   $Z_M^{(2)} \leftarrow g_\xi(X_2)$
4    $\hat{Z}_M^{(1)} \leftarrow \mathrm{SG}(Z_M^{(1)})$,   $\hat{Z}_M^{(2)} \leftarrow \mathrm{SG}(Z_M^{(2)})$
5    $\mathcal{L}_E \leftarrow \frac{1}{2}\left[\mathrm{DV}(Z_E^{(1)}, \hat{Z}_M^{(2)}; \tau) + \mathrm{DV}(Z_E^{(2)}, \hat{Z}_M^{(1)}; \tau)\right]$
6    Update $\theta$ via gradient descent on $\mathcal{L}_E$
7   **end**
  // M-Step: Update $g_\xi$, freeze $f_\theta$ **foreach** *minibatch* $(X_1, X_2) \sim \mathcal{D}$ **do**
8    $Z_E^{(1)} \leftarrow f_\theta(X_1)$,   $Z_E^{(2)} \leftarrow f_\theta(X_2)$
9    $Z_M^{(1)} \leftarrow g_\xi(X_1)$,   $Z_M^{(2)} \leftarrow g_\xi(X_2)$
10    $\hat{Z}_E^{(1)} \leftarrow \mathrm{SG}(Z_E^{(1)})$,   $\hat{Z}_E^{(2)} \leftarrow \mathrm{SG}(Z_E^{(2)})$
11    $\mathcal{L}_M \leftarrow \frac{1}{2}\left[\mathrm{DV}(Z_M^{(1)}, \hat{Z}_E^{(2)}; \tau) + \mathrm{DV}(Z_M^{(2)}, \hat{Z}_E^{(1)}; \tau)\right]$
12    Update $\xi$ via gradient descent on $\mathcal{L}_M$
13   **end**
14 **end**

---

## B.2  JMI CANONICAL FORM TRAINING PROCEDURE

---

**Algorithm 2:** Joint Training Procedure of the JMI Prototype

---

**Input:** Unlabeled dataset $\mathcal{D}$, encoder $f_\theta$, temperature $\tau$, number of epochs $T$
**Output:** Trained encoder parameters $\theta$

1 **for** $t = 1$ **to** $T$ **do**
2   **foreach** *minibatch* $(X_1, X_2) \sim \mathcal{D}$ **do**
3    $Z^{(1)} \leftarrow f_\theta(X_1)$,   $Z^{(2)} \leftarrow f_\theta(X_2)$
4    $\mathcal{L} \leftarrow \frac{1}{2}\left[\mathrm{DV}(Z^{(1)}, Z^{(2)}; \tau) + \mathrm{DV}(Z^{(2)}, Z^{(1)}; \tau)\right]$
5    Update $\theta$ via gradient descent on $\mathcal{L}$
6   **end**
7 **end**

---

# C  METHOD CLASSIFICATION UNDER SDMI/ JMI TAXONOMY

In this section, we present a comprehensive classification of representative SSRL methods under the SDMI/JMI framework as introduced in section 3.

Table 2: Representative SSRL methods and their classification under the SDMI/ JMI taxonomy, with objective types.

| Method | EM/Joint | Objective Type (Explicit MI bound vs. MI Surrogate) | Paradigm |
|---|---|---|---|
| BYOL (Grill et al., 2020) | EM | MI Surrogate | SDMI |
| SimSiam (Chen et al., 2020a) | EM | MI Surrogate | SDMI |
| MoCo-v1/v2/v3 (He et al., 2020; Chen et al., 2020b; 2021) | EM | Explicit MI | SDMI |
| DINO (Caron et al., 2021) | EM | MI Surrogate | SDMI |
| SimCLR (Chen et al., 2020a) | Joint | Explicit MI | JMI |
| Barlow Twins (Zbontar et al., 2021) | Joint | MI Surrogate | JMI |
| VICReg (Bardes et al., 2022) | Joint | MI Surrogate | JMI |
| W-MSE (Ermolov et al., 2021) | Joint | MI Surrogate | JMI |
| SwAV (Caron et al., 2020) | Joint | MI Surrogate | JMI |
| BGRL (Thakoor et al., 2022) | Joint | MI Surrogate | JMI |

# D ADDITIONAL EXPERIMENTAL RESULTS

Supplementary experimental results and analyses support the findings presented in section 4, including controlled experiments on synthetic data and additional ablation studies.

## D.1 CONTROLLED EXPERIMENT

### D.1.1 SDMI DETERMINISTIC FULL-BATCH UPDATES

**Nearest-neighbor angle statistics** To complement fig. 3, we report rotation-invariant NN angle statistics for the final cluster embeddings in table 3. In our setup, on a 3D unit sphere, the ideal separation for five clusters corresponds to a NN gap of $\approx 90°$ (Thomson optimum).

Table 3: NN angle gaps at convergence

| Model | Mean NN Gap (°) | Min NN Gap (°) | Max NN Gap (°) | SD (°) |
|---|---|---|---|---|
| SDMI prototype (Encoder $f_\theta$) | **74.47** | **72.78** | **77.60** | 1.76 |
| SDMI prototype (Encoder $g_\xi$) | 73.77 | 72.16 | 75.06 | **1.32** |
| SimSiam | 40.62 | 28.04 | 71.86 | 16.19 |
| MoCo | 49.64 | 47.90 | 54.07 | 2.29 |
| BYOL | 37.35 | 31.26 | 45.34 | 5.42 |

**Cluster dynamics** In fig. 4, we plot the embeddings of each centroid under both $f_\theta$ and $g_\xi$ encoders at various training iterations (epochs-$1, 5, 20, 70, 100$). As training progresses, the two views of each cluster increasingly align with one another, while the embeddings across different clusters become progressively more separated, indicating that the independently updated encoders learn both consistent and discriminative representations.

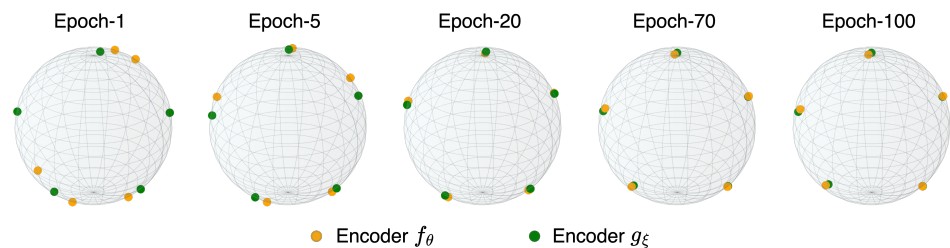

Figure 4: Cluster centers on the unit sphere showing how SDMI prototype encoders progressively separate them

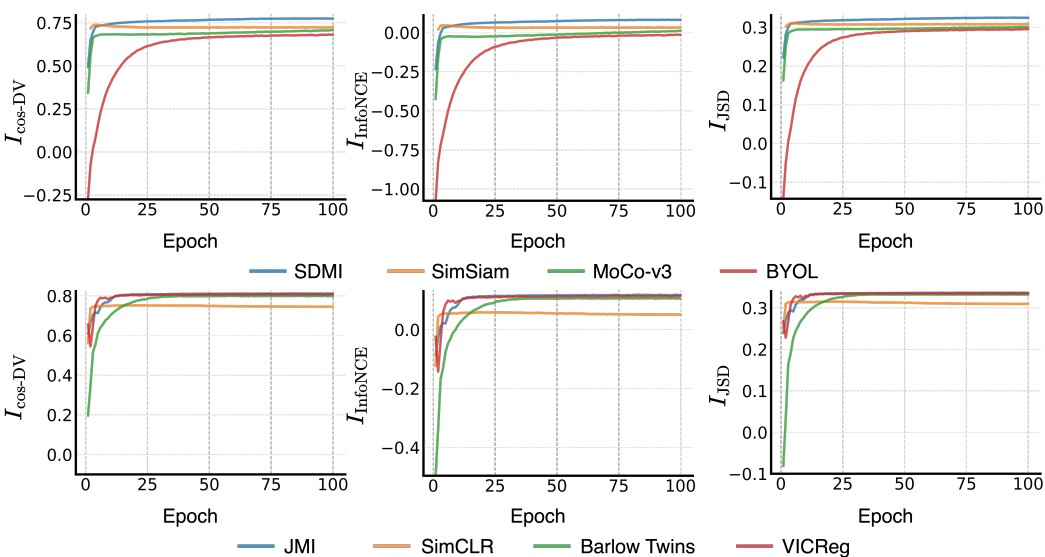

Figure 5: Estimated MI over training using cos–DV, InfoNCE, and JSD bounds for both SDMI methods (top) and JMI methods (bottom). All three estimators show approximately monotonic growth for all methods under both paradigms.

## D.2 SYNTHETIC DATA

To better understand the dynamics of MI maximization without confounding factors from complex image data, we repeat our MI-tracking experiment from section 4 on the same controlled synthetic dataset from section 4.2 using toy models (see section E.4). In this setting, we can isolate the effect of optimization since the ground-truth data distribution is simple and noise is well-characterized. To track MI during training, we compute the same three variational bounds: the cos-DV bound ($I_{\text{cos-DV}}$) from eq. (14), the InfoNCE bound ($I_{\text{InfoNCE}}$) (Oord et al., 2018; Poole et al., 2019), and the JSD bound ($I_{\text{JSD}}$) (Hjelm et al., 2019) using a deterministic setting (i.e., single batch update). At each epoch, we compute all three MI estimates on frozen encoder outputs from a validation set consisting of 2,500 data points. As shown in fig. 5, all three bounds across all methods show similar, near-monotonic MI increase during training.

### D.2.1 SDMI STOCHASTIC MINI-BATCH UPDATES

To examine whether SDMI prototype continues to maximize MI under stochastic optimization, we reran the toy Gaussian mixture experiment using a training batch size of 500 and standard SGD (learning rate 0.5, cosine annealing schedule, 100 epochs). Figure 6 shows the estimated $I_{\text{cos-DV}}$, $I_{\text{InfoNCE}}$, and $I_{\text{JSD}}$ curves for SDMI and the three baseline methods. All approaches continue to show approximately monotonic MI growth despite the use of mini-batches.

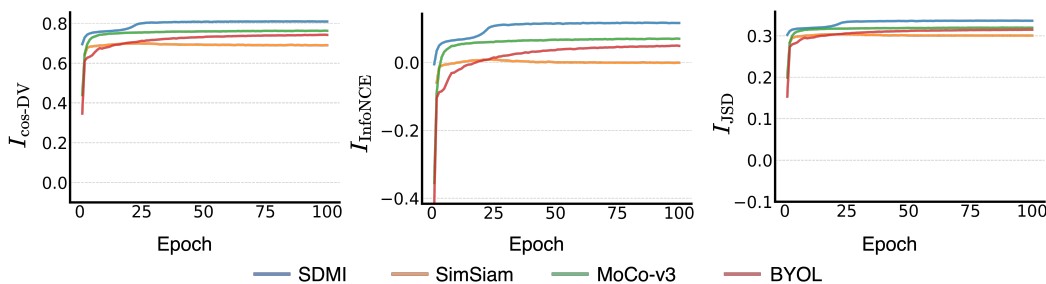

Figure 6: Estimated MI under mini-batch SGD. All methods continue to show monotonic MI growth across three estimators ($I_{\text{cos-DV}}$, $I_{\text{InfoNCE}}$, $I_{\text{JSD}}$).

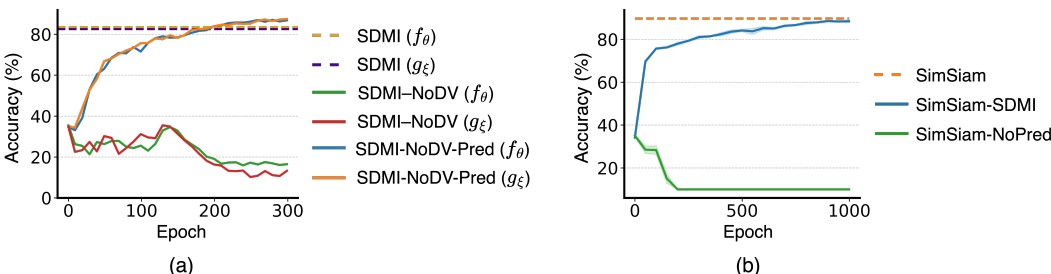

Figure 7: **(a) Linear probing accuracy of SDMI prototype under ablations on CIFAR10.** Removing the marginal term from the DV bound leads to representational collapse in both the $f_\theta$ and $g_\xi$ encoders, denoted as SDMI-NoDV ($f_\theta$) and SDMI-NoDV ($g_\xi$), respectively, despite the use of alternating (EM-style) optimization, confirming the necessity of the marginal regularization term. Remarkably, adding trainable predictors during the E-step and M-step, SDMI-NoDV-Pred ($f_\theta$) and SDMI-NoDV-Pred ($g_\xi$), while still omitting the marginal term, entirely prevents collapse and recovers strong performance. Dashed lines represent baseline accuracy of the pure SDMI prototype with all components intact. **(b) SimSiam variants under different loss functions and predictor configurations on CIFAR10.** Removing the predictor leads to collapse, confirming that SimSiam's cosine loss alone lacks marginal regularization. Replacing the loss with the explicit cos-DV objective (SimSiam-SDMI) restores performance without requiring a predictor.

### D.2.2 PREDICTORS AS DV BOUND MARGINAL SURROGATES

**SDMI prototype** Predictor networks and stop-gradients are widely recognized as essential components in SDMI-based SSRL methods (Chen & He, 2021; Balestriero et al., 2023; Jha et al., 2024; Zhang et al., 2022; Tian et al., 2021; Srinath Halvagal et al., 2023; Shi et al., 2020; Wang et al., 2021). In section 3.2, we showed that stop-gradient enables the EM-style alternating optimization. Zhang et al. (2022) prove that the predictor prevents collapse by decomposing its gradient into center and residual components, showing it induces de-centering and dimensional de-correlation—mechanisms equivalent to those from negative samples in contrastive learning. We complement this understanding with a controlled ablation study on CIFAR10 using linear probing (fig. 7(a)), systematically adding and removing components from the SDMI prototype. Removing the marginal term from eq. (1) results in representational collapse, despite EM-style updates, confirming the necessity of a complete MI objective. More generally, in the absence of the marginal regularization term, encoders converge to trivial solutions.

Remarkably, adding a trainable predictor to the $f_\theta$ encoder during the E-step and to the $g_\xi$ encoder during the M-step, while still omitting the marginal term, entirely prevents collapse. This behavior suggests that predictors function as implicit surrogates for the log-partition (marginal) term, validating our analysis in section 3.2.1 that predictors recover the missing normalization component in MI maximization objective.

**SimSiam** Using SimSiam as an example, we now examine what happens when a non-contrastive method within the SDMI paradigm is given an explicit MI objective. First, we remove the predictor from SimSiam while retaining its alternating update scheme. This variant collapses since minimizing cosine similarity is equivalent to optimizing only the joint term of the DV bound eq. (1), with no marginal correction to prevent trivial solutions. Next, we replace SimSiam's heuristic loss with the cos-DV bound-based loss from eq. (15), while still using a single-encoder alternating schedule but without the predictor. We call this variant SimSiam-SDMI. Remarkably, SimSiam-SDMI not only avoids collapse, but also recovers nearly the original linear probe accuracy (baseline: 89.68±0.35%, SimSiam-SDMI: $88.43 \pm 0.78\%$).

These trends are summarized in fig. 7(b), which shows the performance of all three variants and their associated training behaviors.

# E  IMPLEMENTATION DETAILS

## E.1  COMPUTE RESOURCES

All experiments were conducted on two servers equipped with an NVIDIA RTX 5090 GPU and an NVIDIA RTX 4080 GPU, respectively. The complete set of experiments, including hyperparameter sweeps and additional experiments not included in the paper, took approximately 6 weeks of wall-clock time. For the SDMI prototype, GPU memory (VRAM) usage was around 15 GB, while all other methods required approximately 7 GB.

## E.2  REAL DATA BENCHMARK

**Hyperparameter sweep-1** We conduct a grid search for each model using a ResNet-18 encoder backbone on the CIFAR10/100 datasets. The hyperparameters explored in the initial sweep are summarized in table 4. Models requiring a predictor network (e.g., SimSiam, BYOL, and MoCo-v3) use a fixed 2-layer predictor. For methods that incorporate a temperature parameter (e.g., SDMI, MoCo-v3, JMI and SimCLR), an additional dimension is included in the search space. Momentum-based models such as BYOL and MoCo-v3 use a fixed momentum coefficient of 0.996 for the target encoder. The total number of configurations evaluated per model is shown in table 5.

**Hyperparameter sweep-2** Based on the results from sweep-1, we perform a secondary evaluation for each model on each dataset using a projection layer size of 3. The top-performing configuration (shown in table 6 for each model is then selected and used to train that model for 1000 epochs.

Table 4: **Sweep-1:** Hyperparameter settings and search space used in our grid search

| Parameter | Values | Applies to | Fixed/Varied |
|---|---|---|---|
| Encoder Backbone | ResNet-18 | All models/datasets | Fixed |
| Batch Size | 512 | All models/datasets | Fixed |
| Projection Layers | 2 | All models/datasets | Fixed |
| Prediction Layers | 2 | All models/datasets | Fixed |
| Prediction Dimension | 256 | All models/datasets | Fixed |
| Epochs | 300 | All models/datasets | Fixed |
| Feature Dimension | 2048 | All models/datasets | Fixed |
| Momentum Coefficient | 0.996 | BYOL, MoCo-v3 | Fixed |
| Seed | 7349 | All models/datasets | Fixed |
| Learning Rate | $\{0.01, 0.03, 0.05\}$ | All models (Cosine decay) | Varied |
| Weight Decay | $\{0.0001, 0.0005\}$ | All models | Varied |
| Temperature | $\{0.05, 0.07, 0.1\}$ | SDMI, JMI, MoCo-v3, SimCLR | Varied |
| Projection Dimension | $\{128, 256\}$ | All models | Varied |

Table 5: Grid search configuration counts per model for sweep-1. Configurations are counted per dataset unless otherwise noted.

| Model | Dataset(s) | Temperature Used | Total Configurations |
|---|---|---|---|
| SDMI | CIFAR10, CIFAR100 | Yes | $3 \times 2 \times 3 \times 2 = 36$ |
| SimSiam | CIFAR10, CIFAR100 | No | $3 \times 2 \times 2 = 12$ |
| BYOL | CIFAR10, CIFAR100 | No | $3 \times 2 \times 2 = 12$ |
| MoCo-v3 | CIFAR10, CIFAR100 | Yes | $3 \times 2 \times 3 \times 2 = 36$ |
| JMI | CIFAR10, CIFAR100 | Yes | $3 \times 2 \times 3 \times 2 = 36$ |
| SimCLR | CIFAR10, CIFAR100 | Yes | $3 \times 2 \times 3 \times 2 = 36$ |
| Barlow Twins | CIFAR10, CIFAR100 | No | $3 \times 2 \times 2 = 12$ |
| VICReg | CIFAR10, CIFAR100 | No | $3 \times 2 \times 2 = 12$ |

Table 6: Optimal hyperparameters selected from sweep-2 for each model and dataset. LR = learning rate, WD = weight decay, Temp. = temperature. All models use a ResNet-18 encoder. For VICReg, we fix the similarity, variance, and covariance loss coefficients to 25.0, 25.0, and 1.0, respectively, and set a small numerical stability term $\epsilon = 10^{-4}$. These values remain constant across all runs.

| Model | Dataset | LR | WD | Temp. | Proj. Dim | Proj. Layers | Predictor |
|---|---|---|---|---|---|---|---|
| SDMI | CIFAR10 | 0.05 | 0.0005 | 0.1 | 256 | 3 | No |
| SDMI | CIFAR100 | 0.03 | 0.0005 | 0.1 | 256 | 2 | No |
| SimSiam | CIFAR10 | 0.05 | 0.0005 | – | 256 | 3 | Yes |
| SimSiam | CIFAR100 | 0.05 | 0.0005 | – | 256 | 3 | Yes |
| BYOL | CIFAR10 | 0.05 | 0.0005 | – | 256 | 3 | Yes |
| BYOL | CIFAR100 | 0.05 | 0.0005 | – | 256 | 2 | Yes |
| MoCo-v3 | CIFAR10 | 0.05 | 0.0005 | 0.1 | 256 | 2 | Yes |
| MoCo-v3 | CIFAR100 | 0.05 | 0.0001 | 0.1 | 256 | 2 | Yes |
| JMI | CIFAR10 | 0.05 | 0.0005 | 0.1 | 128 | 2 | No |
| JMI | CIFAR100 | 0.03 | 0.0005 | 0.1 | 256 | 2 | No |
| SimCLR | CIFAR10 | 0.05 | 0.0005 | 0.1 | 128 | 2 | No |
| SimCLR | CIFAR100 | 0.03 | 0.0001 | 0.1 | 128 | 2 | No |
| Barlow Twins | CIFAR10 | 0.05 | 0.0005 | – | 256 | 2 | No |
| Barlow Twins | CIFAR100 | 0.05 | 0.0005 | – | 256 | 2 | No |
| VICReg | CIFAR10 | 0.05 | 0.0005 | – | 256 | 3 | No |
| VICReg | CIFAR100 | 0.03 | 0.0005 | – | 256 | 3 | No |

### E.3 IMAGENET100 AND TINYIMAGENET TRAINING CONFIGURATION

For both the ImageNet100 and TinyImageNet datasets, we adopt a uniform configuration across all models, with only a few dataset-specific adjustments. The settings are summarized in table 7.

Table 7: Hyperparameter settings used for all models trained on ImageNet100 and TinyImageNet.

| Parameter | ImageNet100 | TinyImageNet |
|---|---|---|
| Encoder Backbone | ResNet-50 | ResNet-18 |
| Epochs | 800 | 1000 |
| Warmup Epochs | 10 | 5 |
| Batch Size | 256 | 512 |
| Initial Learning Rate | 0.05 | 0.05 |
| Learning Rate Schedule | Cosine decay | Cosine decay |
| Weight Decay | 0.0001 | 0.0005 |
| Projection Layers | 3 | 3 |
| Projection Dimension | 256 | 256 |
| Feature Dimension | 512 | 512 |

**Note:** For the SDMI model on ImageNet100, the batch size was reduced to **64** (instead of 256) due to the increased memory requirements from the two-encoder setup.

### E.4 TOY MODELS

We implement the toy canonical SDMI and JMI prototypes in section 4.2, along with all benchmarks, each with a dedicated two-layer MLP encoder mapping $\mathbb{R}^2 \to \mathbb{R}^3$. The encoder consists of a linear layer $(2 \to 64)$ with bias, batch normalization, and ReLU activation, followed by a second linear layer $(64 \to 3)$ with batch normalization. The output is normalized to unit $\ell_2$-norm.

