# OpenReview forum: "Self-Supervised Representation Learning As Mutual Information Maximization"
_ICLR.cc/2026/Conference — ICLR 2026 Conference Withdrawn Submission_

### Official Review · Reviewer_NEZp · 2025-10-28

**Soundness:** 2
**Presentation:** 2
**Contribution:** 2
**Rating:** 4
**Confidence:** 4

**Summary:**

The article aims to provide a unified framework for most Self-Supervised Learning (SSL) approaches based on mutual information (MI) maximization using the Donsker–Varadhan (DV) lower bound. The authors employ this bound to define two alternative formulations:

- Self-Distillation MI (SDMI):
The authors consider an SSL setup with two encoders and seek to maximize the DV lower bound on the MI between their representations. This is achieved via a block ascent-based alternating optimization procedure, where one branch is fixed while the other is updated in turn. Methods such as BYOL and MoCo are categorized under this class.

- JointMI:
This variant assumes a single encoder but applies data augmentations to the inputs and maximizes the MI between the resulting representations. The authors claim that the DV-based objective in this setting encompasses both contrastive methods (e.g., InfoNCE) and non-contrastive methods (e.g., Barlow Twins and VICReg).

The article includes numerical experiments on CIFAR-10, CIFAR-100, Tiny ImageNet, and ImageNet-100, reporting linear probing accuracy and visualization results for learned representations.

**Strengths:**

The paper presents an appealing attempt to formulate a unified, information-theoretic framework for SSL, positioning several well-known methods as special cases and providing theoretical interpretations of algorithmic choices such as stop-gradients.

**Weaknesses:**

The novelty of the proposed framework appears limited. As the authors themselves note, prior works have already established connections between information maximization and SSL (including InfoNCE, Barlow Twins, and VICReg). Moreover, the DV bound has previously been employed both for MI estimation (e.g., MINE) and within SSL contexts, as discussed in the uncited reference [a] below. Hence, using MI maximization as an umbrella framework is not a fundamentally new idea.

Since the paper emphasizes MI maximization as a unifying principle for SSL, it should more clearly discuss why maximizing MI between two alternative representations—essentially enforcing general nonlinear dependence—should yield clustered or semantically meaningful features suitable for linear inference. A relevant uncited reference [b] below appears to address this issue by introducing a correlation-based information measure whose maximization promotes linear dependence between representations (further constrained to equality).

The theoretical claims regarding monotonicity and smoothness rely on assumptions that are questionable or even inapplicable when the encoders are implemented as neural networks with ReLU activations.

The DV bound involves an exponential term, which may introduce numerical instability, particularly in high-dimensional representation spaces [a]. This could limit the practical usability of the SDMI formulation.

The experimental results appear weaker than those reported in literature, such as [b] and [c], despite using similar architectures (ResNet-18, ResNet-50) and datasets. The reported accuracies are consistently lower than the baselines in these references.

Refs:

[a] Mroueh, Y., Melnyk, I., Dognin, P., Ross, J., & Sercu, T. (2021). Improved Mutual Information Estimation. AAAI Conference on Artificial Intelligence, 35(10), 9009–9017.
[b] Ozsoy, S., Hamdan, S., Arik, S., Yuret, D., & Erdogan, A. (2022). Self-Supervised Learning with an Information Maximization Criterion. NeurIPS, 35, 35240–35253.
[c] Da Costa, V. G., Fini, E., Nabi, M., Sebe, N., & Ricci, E. (2022). solo-learn: A Library of Self-Supervised Methods for Visual Representation Learning. Journal of Machine Learning Research, 23(56), 1–6.

**Questions:**

1.How does the SSL approach of Mroueh et al. (AAAI 2021) relate to the proposed framework?

2. Are there any numerical stability issues in SDMI, and if so, how do they scale with representation dimensionality?

3. What does MI maximization imply about the clustering properties or downstream task suitability of the learned representations?

4. Regarding Proposition 3.1, how realistic are the concavity and smoothness assumptions, particularly for ResNet-based encoders?

5. Why are the accuracy results lower than what are reported in literature?

---

### Official Review · Reviewer_gJd2 · 2025-10-28

**Soundness:** 3
**Presentation:** 3
**Contribution:** 2
**Rating:** 4
**Confidence:** 4

**Summary:**

Starting from the Donsker- Varadhan (DV) variational form of the mutual information, this paper casts Self supervised techniques as instantiations of the DV lower bound  for a particular choice of encoder and the function class chosen in the DV bound. Authors distinguish between 1)  Self Distillation Mutual information (SDMI) , where two encoders are learned for each view while keeping fix the critic in the DV bound , alternating the optimization with stop gradient on the encoder of the other view. And 2) the joint optimization JMI where the same encoder is learned on both view with fixed critic for the DV.
Authors show for instance that BYOL and SimSiam can be seen as SDMI instances where barloo twins can be derived as taylor series expansion of JMI from the DV bound.
Some experiments on CIFAR 10 and CIFAR 100 and synthetic data explore these objectives from the DV bound.

**Strengths:**

The paper does a nice job revisiting some objective and maps them back to the DV bound and is easy to read and to follow.

**Weaknesses:**

* The main weakness of the paper is that it does not add much to known results on self supervised learning, except maybe that the taylor expansion of DV corresponds to barlow twin, which is a nice result.

* While this variance interpretation as taylor expansion of DV is correct, it can be easily shown that barlow twin is maximizing the mutual information using the chi squared divergence instead of the KL , see for example page 6 , point 2 here for the chi squarred variational form with variance : https://people.lids.mit.edu/yp/homepage/data/LN_fdiv_short.pdf


* DV bounds for estimating mutual information and for self supervised learning have been used before in the literature for instance: mine used https://arxiv.org/pdf/1801.04062 while learning T and  **improved mutual information estimation, AAAI** studies the DV estimator and uses it with learned T and embeddings in self supervised learning. A more thorough review of the literature and how they compare is needed in this study.

**Questions:**

- In equation 15 how do you handle the bias introduced by the log non linearity ?
- would the performance improve if T is also learned along the embedding in JMI as in Improved mutual information estimation, AAAI ?
- What is your overall recommendation and implication of the theory and experiments presented in the paper ? Is there any preference of using SDMI versus JMI?

---

### Official Review · Reviewer_pYZk · 2025-10-31

**Soundness:** 3
**Presentation:** 3
**Contribution:** 2
**Rating:** 4
**Confidence:** 4

**Summary:**

This paper presents a theoretical framework that unifies a wide range of SSL methods through the lens of MI maximization. Starting from DV lower bound on MI, the authors elegantly derive two distinct and principled optimization paradigms: SDMI and JMI. The paper shows that these two paradigms can be used to explain many empirical findings.

**Strengths:**

1. The paper provides a principled derivation that unifies a diverse set of SSL algorithms under two optimization paradigms (SDMI and JMI). This seems more theoretically sound than the traditional contrastive vs. non-contrastive division.

2. The paper provides a formal explanation for why previously "empirically motivated" components (like stop-gradients and predictors) are theoretically essential for practical MI maximization within their respective paradigms.

3. The authors successfully map previous methods into the two paradigms. The demonstration of deriving the Barlow Twins loss from an approximation of the DV bound is interesting.

**Weaknesses:**

1. The paper's foundation relies heavily on using the DV lower bound to analyze the MI. A critical, unaddressed concern is the tightness and fidelity of this bound, then whether the framework optimizes the true mutual information is significantly compromised.

2. The experiments should include larger datasets like ImageNet 1k and more pretraining epochs should be conducted. Also, the experiments show no clear advantage upon the previous SOTA.

3. What are the theoretical advantages of choosing from each of the two paradigms: alternating optimization or joint optimization? There should be more analysis; otherwise, it may just follow from previous SOTA's empirical try, weakening the theoretical contribution.

**Questions:**

See Weakness.

---

### Official Review · Reviewer_4A1U · 2025-11-01

**Soundness:** 2
**Presentation:** 2
**Contribution:** 2
**Rating:** 2
**Confidence:** 4

**Summary:**

This paper proposes a unified theoretical framework for self-supervised representation learning (SSRL) by viewing it through the lens of mutual information (MI) maximization, specifically using the Donsker-Varadhan (DV) bound. The authors derive two optimization paradigms: Self-Distillation MI (SDMI), which relies on alternating updates with stop-gradients (e.g., akin to BYOL, SimSiam, MoCo), and Joint MI (JMI), which allows joint optimization via symmetric architectures (e.g., SimCLR, Barlow Twins, VICReg). They argue that common architectural elements like predictors, stop-gradients, and regularizers are not mere heuristics but emerge naturally from optimizing the MI objective. The paper maps several existing SSRL methods to these paradigms and includes experiments with canonical implementations to validate the framework.

**Strengths:**

1. The work provides a clean, first-principles approach to understanding SSRL by starting from the DV bound on MI. This is a nice way to bridge contrastive and non-contrastive methods under a shared optimization lens, moving beyond prior unifications that focus more on collapse prevention (e.g., Jha et al., 2024) or entropy approximations (Liu et al., 2022).
2. One of the paper's strongest points is showing how elements like stop-gradients in SDMI or regularizers in JMI arise as necessary surrogates for the MI objective.
3. The taxonomy in Section C (appendix) is helpful, categorizing methods like MoCo as SDMI and VICReg as JMI.

**Weaknesses:**

1. MI maximization is already a well-established view in SSRL (e.g., Oord et al., 2018; Poole et al., 2019), and the DV bound isn't new. While the SDMI/JMI split is insightful, it sometimes feels like a rephrasing of existing distinctions (e.g., momentum vs. shared encoders).
This work may have no contribution to the self-supervised field.
2. The second term of the Donsker-Varadhan (DV) bound is a biased estimate. In general, this bound is not used to estimate mutual information. The authors did not adopt any method to reduce bias, making the experiments unable to verify the effectiveness of their method.
3. The literature on information theory and self-supervised learning is very inadequate, and many related works are not mentioned at all.
[1] On Variational Bounds of Mutual Information
[2] Self-supervised Learning from a Multi-view Perspective
[3] Matrix Information Theory for Self-Supervised Learning
[4] Decomposed Mutual Information Estimation for Contrastive Representation Learning
[5] Mveb: Self-supervised learning with multi-view entropy bottleneck
[6] Compressive Visual Representations
[7]Information Flow in Self-Supervised Learning
[8] Rethinking Minimal Sufficient Representation in Contrastive Learning
4. The experiments are solid but not exhaustive. Canonical forms underperform state-of-the-art (e.g., ~70-80% on ImageNet linear probe vs. 90%+ for tuned methods), which is expected but limits claims of "competitive performance.
5. In Section 3.2, how does the implicit M-step in SimSiam (resetting the target) truly mimic coordinate ascent? It seems more like a heuristic—does it guarantee monotonicity like the explicit version?

**Questions:**

See weakness

---

### Note · Authors · 2025-12-01

**Comment:**

We sincerely appreciate the time and effort the reviewers and area chairs dedicated to evaluating our submission. Your comments and suggestions were thoughtful and valuable to us. After careful consideration, we have decided to withdraw the paper from further improvement. We will take the feedback into account as we continue to improve the work.

**Withdrawal Confirmation:**

I have read and agree with the venue's withdrawal policy on behalf of myself and my co-authors.